# Functional genetic encoding of sulfotyrosine in mammalian cells

Xinyuan He[1], Yan Chen[2,5], Daisy Guiza Beltran[3,5], Maia Kelly [2,5], Bin Ma[2], Justin Lawrie[2], Feng Wang [4], Eric Dodds[2], Limei Zhang [3], Jiantao Guo [2✉] & Wei Niu [1✉]

Protein tyrosine O-sulfation (PTS) plays a crucial role in extracellular biomolecular interactions that dictate various cellular processes. It also involves in the development of many human diseases. Regardless of recent progress, our current understanding of PTS is still in its infancy. To promote and facilitate relevant studies, a generally applicable method is needed to enable efficient expression of sulfoproteins with defined sulfation sites in live mammalian cells. Here we report the engineering, in vitro biochemical characterization, structural study, and in vivo functional verification of a tyrosyl-tRNA synthetase mutant for the genetic encoding of sulfotyrosine in mammalian cells. We further apply this chemical biology tool to cell-based studies on the role of a sulfation site in the activation of chemokine receptor CXCR4 by its ligand. Our work will not only facilitate cellular studies of PTS, but also paves the way for economical production of sulfated proteins as therapeutic agents in mammalian systems.

[1] Department of Chemical & Biomolecular Engineering, University of Nebraska-Lincoln, Lincoln, Nebraska 68588, USA. [2] Department of Chemistry, University of Nebraska-Lincoln, Lincoln, Nebraska 68588, USA. [3] Department of Biochemistry, University of Nebraska-Lincoln, Lincoln, Nebraska 68588, USA. [4] Institute of Biophysics, Chinese Academy of Sciences, Beijing, China. [5] These authors contributed equally: Yan Chen, Daisy Guiza Beltran, Maia Kelly. ✉email: jguo4@unl.edu; wniu2@unl.edu

First reported in 1954[1], protein tyrosine O-sulfation (PTS) entails the transfer of a sulfate group from 3′-phosphoadenosine 5′-phosphosulfate (PAPS) to a protein tyrosine residue under the catalysis of membrane-bound tyrosylprotein sulfotransferases (TPSTs) in the trans-Golgi. It is considered the most common type of tyrosine modification in nature and occurs exclusively on secreted and membrane-bound proteins that transit the trans-Golgi network[2–7]. The introduction of negatively charged sulfate groups plays a crucial role in extracellular biomolecular interactions that dictate various cellular processes, including cell adhesion, leukocyte trafficking, hormone activities, and immune responses[3–5,8]. Tyrosine-sulfated proteins (referred hereafter as sulfoproteins) are also involved in the development of various infectious diseases (e.g., AIDS and malaria), cancers, and immune-mediated diseases (e.g., allergy and rheumatoid arthritis)[9–15]. Hence, PTS could emerge as an important therapeutic target for the treatment of human diseases. However, in comparison to protein tyrosine phosphorylation, much of the fundamental questions of PTS remain unanswered. Indeed, the lack of understanding on the role of PTS in mammalian cell biology severely hindered the efforts to unravel potential link(s) between PTS and human diseases.

The key to a successful functional study of a PTS event is the ability to synthesize site-specifically sulfated proteins. Although sulfoproteins or sulfopeptides (fused to secreted proteins) can be synthesized in eukaryotic cells with normal[16–19] or enhanced[11,20,21] TPST activities, this method lacks the resolution that is required to investigate a specific tyrosine O-sulfation event within a protein containing multiple sulfotyrosine (sTyr) residues. As such, chemically synthesized sulfopeptides[22–25], became an important tool to model the structural and functional role of PTS[5]. Regardless of recent progress[5,26], the preparation of sTyr-containing peptides is still challenging. Furthermore, sulfopeptides may not faithfully resemble the native conformation and biological function of full-length sulfoproteins. To meet above challenges, the genetic incorporation of sTyr into proteins in response to amber nonsense codon (TAG) was reported[27]. This method enables the synthesis of site-specifically sulfated proteins in Escherichia coli (E. coli) using engineered Methanocaldococcus jannaschii tRNA ($Mj$tRNA$^{Tyr}$) and its cognate, sTyr-specific tyrosyl-tRNA synthetase (sTyrRS) pair. However, the reported $Mj$tRNA$^{Tyr}$-sTyrRS pair is limited to E. coli host and cannot be used in mammalian cells where protein sulfation happens and plays important biological roles. While our manuscript was under preparation, Chatterjee et al. published an elegant work on the genetic incorporation of sTyr in mammalian cells[28].

Here we report the synthesis of site-specifically sulfated proteins in both live yeast and mammalian cells through the genetic incorporation of sTyr using an engineered tyrosyl-tRNA synthetase derived from E. coli ($Ec$TyrRS). Unlike the recently published work by Chatterjee et al.[28], our engineering of $Ec$TyrRS is conducted by using yeast as the selection host. Although E. coli is a more widely adopted platform in the directed evolution of aminoacyl-tRNA synthetases (aaRSs), yeast has the translational machinery that is more similar to that of the mammalian cells. Besides an indirect fluorescence-based assay in live cells and mass spectrometry analysis, we also conduct in vitro enzyme assays that provide direct activity profiles of evolved sTyrRSs. In order to understand the shifted substrate specificity, we obtain a crystal structure of the evolved sTyrRS in complex with sTyr, which also shed lights on future engineering of the enzyme for substrates with similar physicochemical properties. Finally, we perform a cellular and functional study of a co-translationally synthesized sulfoprotein in live mammalian cells. The study focuses on the role of PTS in chemokine signaling, which is central to chronic inflammatory conditions and is associated with the development

of many human diseases. In addition to fundamental biochemical studies, this work also opens a door to the production of sulfoproteins as therapeutic agents in eukaryotes.

## Results

**Library construction and selection**. The crystal structure of E. coli tyrosyl-tRNA synthetase ($Ec$TyrRS) in complex with L-tyrosine (PDB ID: 1X8X)[29] was chosen in an initial protein modeling effort to identify key residues that may lead to shifted substrate specificity. We first rebuilt the ligand in the active site by adding a sulfate group to the hydroxyl group of tyrosine. The simulation revealed that four active site residues (Y37, L71, W129, and D182; Supplementary Fig. 1a) engaged in unfavorable interactions with the new ligand. These four residues were selected for the construction of an $Ec$TyrRS mutant library. Amino acid sequence at each of the four positions were fully randomized using the NNK codon (N=A, C, T, or G; K=T or G; 32 variants at nucleotide level) in mutagenic overlap PCR reactions. Full-length mutant $Ec$TyrRS genes were inserted behind an ADH1 promoter in a p$Ec$TyrRS-lib vector, which also encoded an amber suppressor tRNA derived from E. coli tyrosyl-tRNA ($Ec$tRNA$_{CUA}$). Using the yeast homologous recombination cloning method[30], a library of $1 \times 10^7$ clones was obtained, which represented an over 99% coverage of the theoretical library diversity[31]. The quality of the obtained library was verified by DNA sequencing (Supplementary Fig. 1b).

Selection of sTyr-specific $Ec$TyrRS variants adapted an approach that was previously established in Saccharomyces cerevisiae (Supplementary Fig. 1c)[32–34]. The selection host MaV203 contained chromosomal modifications that allowed the expression of a reporter enzyme, URA3, under the regulation of the transcriptional activator GAL4, which was encoded by a mutant gene containing amber codons at positions 44 and 110. Plasmid DNA of the $Ec$TyrRS library was transformed into MaV203 together with the GAL4-encoding plasmid, pGADGAL4[32]. In a positive selection, yeast cells were cultured in the presence of sTyr (1 mM). Effective amber suppression by $Ec$TyrRS mutants that can charge the amber suppressor $Ec$tRNA$_{CUA}$ with sTyr or any one of the 20 natural amino acids enabled the growth of MaV203 host on media without uracil supplementation. In a negative selection, yeast cells were cultured in the absence of sTyr. Undesirable aminoacylation of $Ec$tRNA$_{CUA}$ by $Ec$TyrRS mutants with natural amino acids led to the conversion of exogenously added 5-fluoroorotic acid (5-FOA, 0.1%) into a cytotoxic compound and caused cell death. A total of $1 \times 10^7$ library cells was subjected to positive selection. Survived cells were then applied to the negative selection. Single colonies resulting from the two rounds of selection were further replicated on screening plates to verify sTyr-dependent growth in the absence of uracil and good cell fitness in the presence of 5-FOA. Four clones were confirmed to have the correct phenotype. DNA sequencing revealed two unique mutants (c1 and c2 in Fig. 1a). Positions Leu71 and Asp182 showed a complete convergence to Val and Gly, respectively, in both mutants. While Tyr37 did not change, position Trp129 either maintained the wild-type sequence or changed to Phe.

**Characterization of candidate sTyr tRNA synthetases (sTyrRSs)**. We first evaluated the activity of obtained sTyrRS candidates by measuring amber suppression efficiency in yeast cells. Isolated library plasmids encoding c1 or c2 were transformed into host strain MaV203 together with a reporter plasmid, pyeGFP-N149TAG, which harbored a yeast enhanced green fluorescent protein gene (yeGFP) with an amber mutation at the N149 position. The fluorescence intensity of yeGFP was compared when cells were cultured without or with sTyr (Fig. 1b). The two sTyrRS variants led to similar levels of background fluorescence

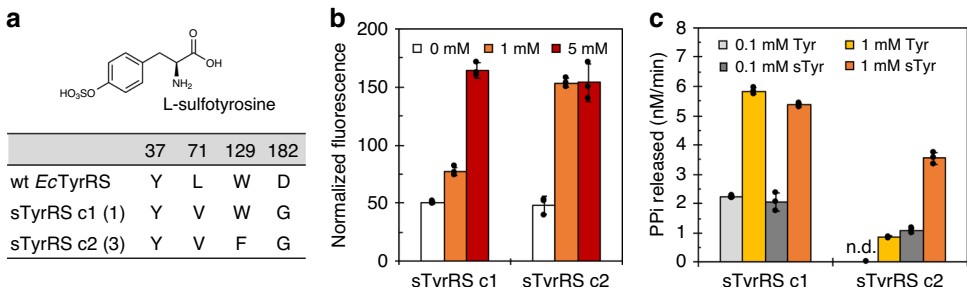

**Fig. 1 Characterization of sTyrRS candidates. a** Mutations in the evolved sTyrRS variants. Numbers in parentheses represent the occurrence of the variant from the selection. **b** Fluorescence readings of yeast cells expressing the evolved sTyrRSs and an yeGFP mutant that contained an amber mutation at position N149. sTyr was included in culture media at 0 mM (open bar), 1 mM (filled orange bar), or 5 mM (filled red bar). Fluorescence intensity was normalized to cell growth. **c** Initial velocities of pyrophosphate release by sTyrRS variants in the presence of Tyr and sTyr. Tyr was included in assays at 0.1 mM (filled gray bar) or 1 mM (filled yellow bar). sTyr was included in assays at 0.1 mM (filled dark gray bar) or 1 mM (filled orange bar). Assays were carried out in the presence of 10 µM recombinant $Ec$tRNA$_{CUA}$. n.d., not detected. **b, c** Data are plotted as the mean ± standard deviation (s.d.) from $n = 3$ independent experiments. **b, c** Source data are provided as a Source data file.

in the absence of sTyr and comparable yeGFP expression in the presence of 5 mM sTyr. The suppression of the amber codon in pyeGFP-N149TAG suggested that both c1 and c2 were able to aminoacylate $Ec$tRNA$_{CUA}$ with sTyr in yeast. Furthermore, fluorescence in cells expressing c2 reached saturation at 1 mM of sTyr (Fig. 1b), which indicated better substrate recognition.

To further compare the biochemical properties of the two sTyrRS variants, we characterized them by in vitro assays. Each enzyme was expressed and purified as a fusion protein with a 6xHis tag on the C terminus. Such modification has previously been reported to have a negligible effect on the function of the wild-type $Ec$TyrRS[35]. The aminoacyl-tRNA synthetase activities were measured by monitoring the release of pyrophosphate in the amino acid activation step of the tRNA acylation reaction[36]. The assays were carried out in the presence of Tyr or sTyr as the substrate (Fig. 1c). The c1 variant showed similar level of activities on the two substrates at both 0.1 mM and 1 mM. Meanwhile, the c2 variant was more specific for sTyr. At the same concentration of 0.1 mM, the enzyme had no detectable activity on Tyr, but was active on sTyr. It also showed a fourfold higher activity on sTyr than Tyr when the substrate concentrations were increased to 1 mM. Overall, the c2 variant had better substrate specificity and maintained similar level of activities towards sTyr as c1. It was chosen as the sTyrRS in following studies.

**Evaluation of sTyrRS in mammalian cells**. We examined the evolved sTyrRS for its ability to incorporate sTyr in mammalian cells using a two-plasmid method. The sTyrRS-encoding gene was inserted behind a non-regulated CMV promoter on plasmid psTyrRS, which has one copy of the amber suppressor tRNA under the control of a human U6 promoter. The reporter plasmid, pEGFP[37], contains an EGFP gene with an amber mutation at the Y40 position and DNA sequence that encoded a C-terminus 6xHis tag. The selective incorporation of sTyr into proteins was tested in 293 T cells that were transiently transfected with plasmids psTyrRS and pEGFP. Following 24 h of incubation in media without and with sTyr (1 mM), the amber suppression level was studied by visualizing the expression of fluorescent protein under confocal microscope (Fig. 2a). Significant amount of EGFP was only expressed in cells supplemented with sTyr, while minimal background was observed without sTyr (Fig. 2a). In addition to cell imaging, flow cytometry was used to quantify the amber suppression efficiency based on the fluorescence intensity (Supplementary Fig. 2). As shown in Fig. 2b, when the suppression efficiency of the wild-type $Ec$TyrRS was assigned as

100%, the efficiency of sTyrRS is about 68%, which is significantly higher ($P < 0.01$) than a well-established $Ec$TyrRS mutant, AzFRS (for the incorporation of p-azido-L-phenylalanine; AzF[38,39]). On the other hand, the evolved sTyrRS did show a higher background signal than AzFRS when noncanonical amino acid (ncAA) was not available (Fig. 2b). We also conducted protein expression of an EGFP mutant that contains two amber codons in its encoding gene at positions Y40 and N150 (on plasmid pEGFP-double). As shown in Supplementary Fig. 3, EGFP fluorescence was only detected when 1 mM sTyr was included in the cell culture media. The expression level of EGFP-double (Supplementary Fig. 3) was apparently lower than that of EGFP containing one amber codon (Fig. 2a).

Finally, the fidelity of the evolved sTyrRS was quantified by mass spectrometry. Since the sulfoester bond in sTyr is labile and can readily decompose to yield Tyr under mass spectrometry conditions[5], it would be impossible to directly analyze the ratio of sTyr versus mis-incorporated Tyr in the target protein. To solve this problem, we synthesized and used di-deuterium labeled sTyr (sTyr-D$_2$; Supplementary Fig. 4a) in the incorporation study. To reduce the cost of this experiment, we used a mixture of sTyr-D$_2$ and unlabeled sTyr instead of pure sTyr-D$_2$. The mixture contained sTyr-D$_2$ at 24.2 ± 0.3% (mol%) by mass spectrometry quantification. To conduct the analysis, the EGFP-Tyr40TAG was expressed in 293T cells in the presence of the sTyr-D$_2$/sTyr mixture at 1 mM. Since the cell culture media (DMEM with 10% FBS) contains at least 0.4 mM Tyr (a component of DMEM), we should see a significant dilution of the deuterium label at position 40 of EGFP if the evolved sTyrRS incorporates Tyr at a substantial level. The mutant protein was purified by Ni affinity resin followed by SDS-PAGE (Supplementary Fig. 4b), then subjected to trypsin digest. The resulting peptides were analyzed by a Synapt G2S HDMS quadrupole time of flight (Q-TOF) hybrid mass spectrometer. The peptide fragment that contained the position of incorporation showed the percentage of Tyr-D$_2$ at position 40 as 24.3 ± 1.3%, which is in close agreement with the percentage of di-deuterium labeled sTyr supplemented to the cell culture media (Supplementary Fig. 5). The data excluded misincorporation of unlabeled Tyr from the cell culture media at position 40 of EGFP in the presence of sTyr. Otherwise, it would lead to the dilution effect and lower the percentage of deuterium labeled peptide. While some background incorporation of Tyr was observed in the absence of sTyr (Fig. 2a), the above studies demonstrate that the evolved sTyrRS can aminoacylate amber suppresser tRNA with high efficiency and fidelity in the presence of sTyr in mammalian cells.

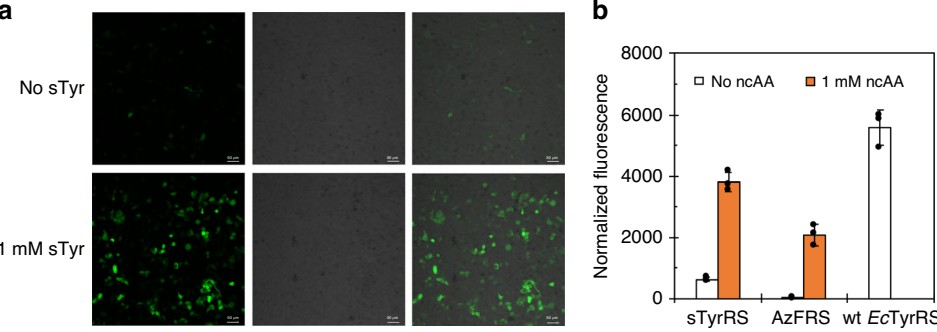

**Fig. 2 Evaluation of sTyrRS in mammalian cells. a** Confocal images of 293T cells expressing the evolved sTyrRS and an EGFP mutant that contained an amber mutation at position Tyr40. The left panel shows fluorescent images, the middle panel shows brightfield images, and the right panel shows composite images of bright-field and fluorescent images. Scale bars, 50 μm. **b** Flow cytometry analyses of 293T cells expressing the evolved sTyrRS, AzFRS, and wild-type *Ec*TyrRS, each with the EGFP-Y40TAG mutant. The normalized fluorescence was calculated by multiplying the mean fluorescence intensity by the percentage of fluorescent cells in flow cytometry analyses in the absence (open bar) or presence (filled orange bar) of ncAA. **b** Data are plotted as the mean ± s.d. from n = 3 independent experiments. The raw data of flow cytometry experiments are shown in Supplementary Fig. 2. **b** Source data are provided as a Source data file.

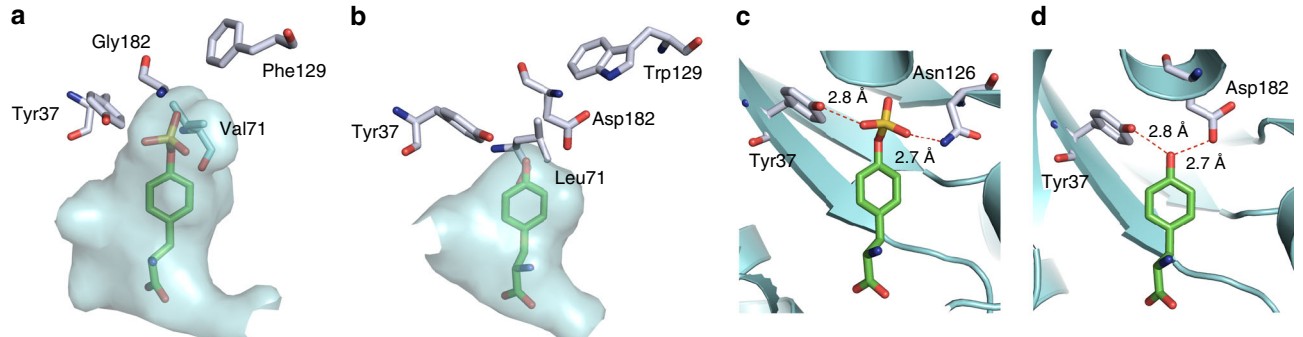

**Fig. 3 Amino acid binding pocket of sTyrRS and wild-type *Ec*TyrRS. a–b** Active site close-ups of the evolved sTyrRS (**a**, PBD ID: 6WN2) and *Ec*TyrRS (**b**, PDB ID: 1X8X). The active site surfaces are in cyan. Four residues included in library construction are labeled. **c–d** Recognition of ligands by sTyrRS (**c**) and the wild-type *Ec*TyrRS (**d**). Carbon atoms in ligands are shown in green. Carbon atoms in proteins are shown in silver. All oxygen atoms are in red, nitrogen atoms are in blue.

**Structural insights into the evolved sTyrRS**. To understand key changes that shift the substrate specificity of the evolved sTyrRS, we conducted structural study of its amino acid activation domain (a.a. 1-322). The domain was overexpressed and purified as a fusion protein with an N-terminal 6xHis-SUMO tag, which was removed by treatment with the SUMO protease prior to crystallization. We obtained and refined the crystal structure of the evolved sTyrRS in complex with sTyr at 1.78 Å resolution. The protein formed a homodimer, the same as the wild-type *Ec*TyrRS (Supplementary Fig. 6a)[29]. In comparison to the structure of the wild-type *Ec*TyrRS in complex with L-Tyr (PDB ID: 1X8X), the introduction of the L71V and D182G mutations in the evolved sTyrRS led to an expanded active site to accommodate the larger side chain of sTyr (Fig. 3a, b). In addition, the D182G mutation eliminates the potential repulsion between the negatively charged carboxylate group of Asp and the sulfate group of sTyr. This mutation also contributed to the lowered affinity of the evolved sTyrRS towards Tyr, due to the abolishment of a stabilizing hydrogen bond (Fig. 3d). A third mutation, W129F, contributes to the substantially lower background activity of the c2 variant towards Tyr. This is likely due to the smaller size of the Phe side chain, which further contributes to the enlarged active site (Fig. 3a). A close inspection of ligand binding revealed that the phenol ring of the sTyr tilted around 26° away from that of the Tyr (Supplementary Fig. 6f). This movement enabled the

formation of a hydrogen bond between the sulfate group and the hydroxyl group of Tyr37 residue (Fig. 3c). Since this Tyr is also critical for the binding of Tyr substrate in the active site of the wild-type *Ec*TyrRS (Fig. 3d), its retention likely played a major role in the background activity of the evolved sTyrRS towards Tyr when sTyr was not available. Another key interaction that anchors the sulfate group is a hydrogen bond with the side chain of Asn126 (Fig. 3c), which does not participate in substrate binding in the wild-type *Ec*TyrRS. Finally, the three mutations in the evolved sTyrRS did not interrupt the three hydrogen bonds that are essential for recognizing the α-amino group of the substrate (Supplementary Fig. 6d and 6e).

**Functional incorporation of sTyr in CXCR4 protein by sTyrRS**. To demonstrate that sTyr can be functionally and selectively incorporated into naturally sulfated positions of target proteins in live mammalian cells, we conducted studies of chemokine receptor CXCR4. The activation of CXCR4 by its cognate chemokine ligand, stromal cell-derived factor 1 (SDF-1; also known as CXCL12), triggers multiple downstream signal transduction pathways that regulate intracellular calcium flux, chemotaxis, transcription, and cell survival. It is known that sulfation of tyrosine residues (Y7, Y12, and Y21, Supplementary Fig. 7a) near the N-terminus of CXCR4 is required

for optimal binding with SDF-1. According to previously reports, the relative importance of the three sulfation sites of CXCR4 for its ligand binding and activation is Tyr21 > Tyr12 > Tyr7[40,41]. Therefore, we decided to focus on the Tyr21 site of CXCR4 in this work.

The incorporation of sTyr at the 21 position of CXCR4 was examined using plasmids psTyrRS and pCXCR4(21TAG)-EGFP, which encoded a CXCR4 mutant as a fusion protein with EGFP on its C terminus. The suppression of the amber mutation in CXCR4 led to the expression of EGFP in 293 T cells of which the fluorescence was monitored by flow cytometry. In the presence of sTyr, a twofold increase in fluorescence was observed, which was 43.5% of the suppression efficiency when the wild-type $Ec$TyrRS was used (Supplementary Fig. 7b). To further confirm that sTyr was specifically incorporated at position 21, we also conducted western blot analyses. As a membrane protein, CXCR4 showed anomalous migration in SDS-PAGE[42]. To circumvent this issue, we fused the DNA sequence that encodes the first 38 residues of CXCR4 with the EGFP-encoding gene containing a 6xHis tag on its C terminus. The resulting pCXCR4(N38-FF 21TAG)-EGFP plasmid also included Y7F, Y12F and an amber mutation at the Tyr21 position. The Y7F and Y12F mutations eliminate any possibility of TPST-catalyzed sulfation at these two sites and makes the position 21 the only site that contains sTyr residue through the sTyrRS-mediated incorporation, which simplifies the interpretation of western blot experiments with the anti-sTyr antibody.

Probing lysates of 293T cells transfected with plasmids pCXCR4(N38-FF 21TAG)-EGFP and psTyrRS with anti-histidine tag antibody showed the higher expression level of a protein corresponding to the size of CXCR4(N38 FF 21sTyr)-EGFP when sTyr (1 mM) was provided to cells (Fig. 4a, Supplementary Fig. 7c). Protein with His-tag was partially purified using Ni resin (Supplementary Fig. 7d) and further subject to western blot using both anti-histidine tag and anti-sTyr antibodies (Fig. 4b). The sample from cells cultured with sTyr showed positive signals with both antibodies. In contrast, the sample from cells cultured without sTyr was only positive with the anti-histidine antibody (Fig. 4b). As controls, purified CXCR4 (N38)-EGFP and CXCR4(N38-FFF)-EGFP, which had Tyr to Phe mutations at all three positions, were also only detected by anti-histidine tag antibody. Since the CXCR4(N38)-EGFP variants were cytosolic, they did not transit through trans-Golgi network where PTSTs reside and were not sulfated by the endogenous

protein sulfation machinery. Results of the above western blot analyses confirmed that sTyr was successfully incorporated in response to the amber codon at position 21.

Next, we examined the function of translationally incorporated sTyr at position 21 of CXCR4 using the calcium mobilization assay. To eliminate the background signal caused by the expression of endogenous CXCR4 in 293T cell line, we generated a CXCR4-knockout cell line, 293T-ΔCXCR4 (C4), using CRISPR/Cas method (see details in Supplemental Information)[43]. An approximate 90% reduction of signal in calcium mobilization assay was observed when C4 was treated with SDF-1 (Fig. 4c). The signal in C4 was restored when the wild-type CXCR4 was expressed from plasmid pCXCR4 and reduced back to near basal level when a CXCR4-FFF mutant, which had Y7F, Y12F, and Y21F mutations, was expressed (Fig. 4c). The tyrosine to phenylalanine mutation has been widely used to remove sulfation sites with minimal structural perturbation to target proteins[9,11,44,45]. To specifically probe the functional incorporation of sTyr at position 21 in CXCR4, calcium mobilization in C4 cells expressing CXCR4(FF TAG) (Y7F, Y12F, and Y21TAG) was studied. As shown in Fig. 4c, SDF-1-triggered calcium mobilization was observed when CXCR4(FF TAG) was co-expressed with the evolved sTyrRS in the presence of 1 mM sTyr. On the other hand, over twofold lower and near basal level calcium mobilization was detected when CXCR4(FF TAG) was expressed in the absence of the evolved sTyrRS ($p < 0.05$). The above results validated a previous finding[40,41] that sulfation at Tyr21 position of CXCR4 is important for the activation of this receptor by its chemokine ligand. Overall, our work demonstrated that genetically incorporated sTyr can reconstitute the function of CXCR4 and is suitable for cellular studies.

## Discussion

By using yeast as the selection host, we have identified $Ec$TyrRS mutants that were able to genetically encode sTyr in both yeast and mammalian cells. To overcome the problem of low transformation efficiency with circular DNA, linear DNA fragments were transformed into yeast cells and assembled into $Ec$TyrRS library through the host's homologous recombination machinery. We could regularly obtain libraries of $10^7$ diversity, which is comparable to the capacity of the commonly used E. coli platform. This would facilitate current efforts on engineering

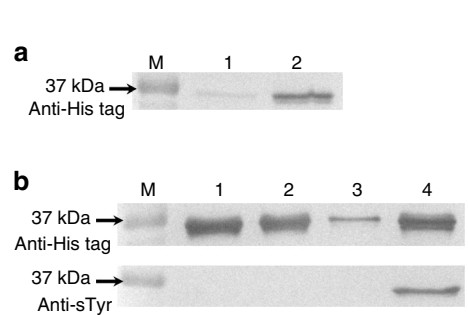

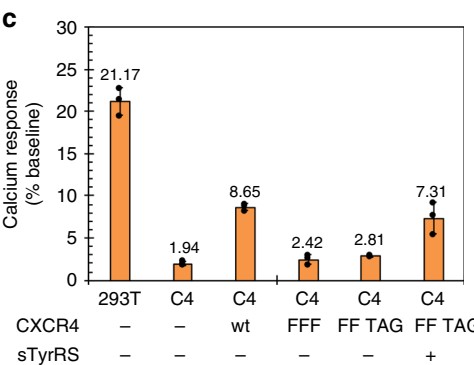

**Fig. 4 Functional incorporation of sTyr in mammalian cells. a** Western blot analysis of CXCR4(N38-21TAG)-EGFP expression in the absence (lane 1) and presence of sTyr (1 mM, lane 2) in 293T cells with psTyrRS. **b** Western blot analysis of partially purified CXCR4(N38)-EGFP variants. Lane 1, CXCR4(N38)-EGFP; lane 2, CXCR4(N38-FFF)-EGFP; lane 3, CXCR4(N38-FF 21TAG)-EGFP in the absence of sTyr; lane 4, CXCR4(N38-FF 21TAG)-EGFP in the presence of sTyr (1 mM); **c** Calcium mobilization assay. wt, wild-type CXCR4; FFF, CXCR4(FFF) mutant; FF TAG, CXCR4(FF 21TAG) mutant. Lane M in both (**a**) and (**b**) is molecular weight marker. **c** Data are plotted as the mean ± s.d. from $n = 3$ independent experiments. **a–c** Source data are provided as a Source data file.

aaRSs for genetic incorporation of ncAAs into proteins in eukaryotic cells.

The evolved sTyrRS displayed good efficiency in comparison to its parental wild-type EcTyrRS and to a well-established AzFRS. While a side-by-side evaluation has not been conducted with evolved sTyrRSs from this and Chatterjee's work[28], comparable activities are expected since they have similar mutations, including Leu71Val and Asp182Gly. The Chatterjee's sTyrRS does not have the Trp129Phe mutation as the sTyrRS in this report, but does contain a unique Leu186Met mutation. Residue Leu186 sits on the same α helix as residue Asp182, but is further away from the ligand in the wild-type EcTyrRS (Supplementary Fig. 8a). In the expanded substrate pocket of our mutant sTyrRS, Leu186 resides on the active site surface and mutation at this site, similar to our Trp129Phe mutation, may fine tune the sTyr binding pocket (Supplementary Fig. 8b). Incorporation of the Leu186Met or other mutations at this position of our sTyrRS mutant may lead to further improvement of properties. The evolved sTyrRS displayed certain level of background. However, mis-incorporation of Tyr was not observed in the presence of sTyr, which suggests that the evolved sTyrRS is more active on sTyr than Tyr. This notion was confirmed by the results of in vitro enzyme assays. Our structural study showed that Tyr37 could form a hydrogen bond with either sTyr or Tyr. It therefore can potentially contribute to the background activity of the evolved sTyrRS towards Tyr. Further mutagenesis that weakens hydrogen bonding between Tyr37 and the undesirable Tyr substrate can likely lead to sTyrRS variants with higher specificity. We are currently conducting molecular dynamics simulations using the obtained sTyrRS structure to identify potential sites for modification. Our findings will be reported in due course.

In a proof of concept study using the CXCR4/SDF-1 axis as the model system, we demonstrated that the genetically incorporated sTyr could be readily used to study the role of sTyr in the activation of chemokine receptors in live mammalian cells. While a single sulfation site (sTyr21) was the focus of this work to demonstrate the utility of genetic encoding of sTyr in live-cell studies, it is possible to investigate multiple sulfation sites simultaneously in the future. However, caution is recommended in such studies since the efficiency of genetic incorporation of sTyr is likely decreased due to the need to suppress multiple amber codons within the same transcript. Since many G protein-coupled receptors, including chemokine receptors, are sulfated, it is crucial to understand whether and how sulfation events affect receptor-ligand interactions. The system established in this work is a useful tool to facilitate the study of protein sulfation-associated eukaryotic signaling processes and the development of diagnostic and therapeutic agents targeting protein sulfation. Furthermore, the genetic incorporation of sTyr is expected to enable facile productions of therapeutic proteins containing tyrosine sulfation, which has been reported as a key post-translational modification to boost the efficacy[46].

## Methods

**Reagents and media**. All commercial chemicals are of reagent grade or higher. All solutions were prepared in deionized water that was further treated by Barnstead Nanopure® ultrapure water purification system (Thermo Fisher Scientific). Preparation of LB medium and YPD medium followed the reported recipe. Yeast selection media contained yeast nitrogen base without amino acids (Difco™, 6.7 g/L), glucose (10 g/L), and appropriate drop out (DO) supplements (Clonetech). The pH values of all media were adjusted to 7.0. Agar plates were prepared by the addition of Difco agar (15 g/L) to the liquid medium. Antibiotics were added where appropriate to the following final concentrations: kanamycin (50 mg/L), ampicillin (100 mg/L). IPTG stock solution was added as necessary to the indicated final concentration. Solutions of antibiotics and IPTG were filtered through 0.22 μm sterile membrane filters. Sulfotyrosine was synthesized by following our previously published procedure[47]. In brief, trifluoroacetic acid (50 mL) and L-tyrosine (10 g, 55.6 mmol) were added into a dry round-bottom flask with a magnetic stir bar at

−10 °C. While stirring under nitrogen protection, 5 mL (75.2 mmol) of chlorosulfonic acid was added over 2 min. The reaction was stirred for an additional 5 min, quenched by the addition of ethanol (3 mL), and stirred for another 2 min at room temperature. Sulfotyrosine was precipitated with ether (175 mL), filtered, and washed three times with ether (75 mL per wash). The product was dried under vacuum to remove residual ether.

**Mammalian cell experiments**. Mammalian cells were maintained in DMEM (Gibco™) media supplemented with 10% FBS (v/v) at 37 °C in a humidified atmosphere of 5% $CO_2$ (v/v). Transfection was conducted at 70–80% cell confluency using Lipofectamine 2000 (Life Technologies) according to the manufacturer's protocol. For purification of protein with 6xHis tag, cells from a T-25 or T-75 flask were detached and lysed in RIPA lysis and extraction buffer (Thermo Scientific) with protease inhibitor cocktail (EDTA-free, Thermo Scientific). Following centrifugation at 21,000 × g for 30 min at 4 °C, the supernatant was collected and dialyzed into purification buffer, which contained potassium phosphate (25 mM, pH 7.5), NaCl (200 mM), and imidazole (10 mM). The tagged protein was purified using Ni Sepharose 6 Fast Flow resin (GE Healthcare) by following the manufacturer's protocol.

**Biochemical analysis**. Protein concentrations were determined by using the Quick Start™ Bradford protein assay kit (Bio-Rad Laboratories). Western blot experiments started with sample separation by SDS-PAGE. Proteins were then transferred onto a nitrocellulose membrane using mini trans-blot cell (Bio-Rad Laboratories) at 100 V for 1 h. The membranes were blocked overnight at 4 °C, then incubated with primary and secondary antibodies for 1 h each: mouse anti-6xHis tag monoclonal antibody (1:3000, MCA1396, Bio-Rad Laboratories), mouse anti-sulfotyrosine antibody clone sulfo-1C-A2 (1:1000, 05-1100, MilliporeSigma), and HRP conjugated goat anti-mouse IgG (1:1000, 1706516, BioRad Laboratories). Blots were developed using Opti-4CN detection kit (BioRad Laboratories) and imaged using Gel Doc XR+ system. A Biotek Synergy HTX plate reader was used in absorbance, fluorescence, and kinetics measurements.

**Strains and plasmids**. E. coli GeneHogs® (Thermo Fisher Scientific) was used for routine cloning and plasmid propagation. Plasmid construction was performed using Gibson Assembly method[48]. E. coli BL21(DE3) (New England Biolabs) was used for protein expression and purification. S. cerevisiae strain MaV203 (MATα; leu2-3,112; trp1-109; his3Δ200; ade2-101; cyh2R; cyh1R; gal4Δ; gal80Δ; GAL1::lacZ; HIS3$_{UASGAL1}$::HIS3@LYS2; SPAL10$_{UASGAL1}$::URA3) (Thermo Fisher Scientific) was used for yeast library construction and selection. Standard protocols were followed for the purification and analysis of plasmid DNA[49]. PCR amplifications were carried out using KOD Hot Start DNA polymerase (MilliporeSigma) by following the manufacturer's protocol. Other molecular cloning reagents were purchased from New England Biolabs. Primer synthesis and DNA sequencing services were provided by Eurofins MWG Operon. Primers used in this study are listed in Supplementary Table 1. Plasmids used and constructed in this study are listed in Supplementary Table 2. All plasmids are available from corresponding authors upon reasonable request.

**Plasmid pyeGFP-N149TAG**. This 7.3-kb plasmid is a pGAD-derived plasmid that contains a mutant yeast enhanced green fluorescent protein (yeGFP) gene behind the yeast glyceraldehyde-3-phosphate dehydrogenase promoter (TDH3). Promoter TDH3 was amplified from the genomic DNA of S. cerevisiae strain BY4742. Gene of yeGFP was amplified as two fragments from plasmid pDZ276[50] (Addgene 35194). The N149TAG mutation was introduced using primers for the amplification of the second fragment. Obtained PCR products were purified by DNA electrophoresis followed by gel isolation, then assembled into pGAD vector that was prepared by removing a mutant GAL4 gene from pGADGAL4[32] by SphI and AgeI digestion.

**Plasmid pET30b-sTyrRSs**. These 6.7-kb plasmids are pET30b-derived plasmids that encode candidate sTyrRSs with a C-terminus 6xHis tag. DNA fragments encoding candidate sTyrRSs were amplified without the stop codon using isolated library plasmids as templates. PCR products were assembled into pET30b (Novagen) vector that was linearized by digestion with NdeI and XhoI.

**Plasmid psTyrRS and pEcTyrRS**. These 7.4-kb plasmids are pcDNA3.1-derived plasmids that contain one copy of the amber suppressor B. subtilis tyrosyl-tRNA behind a human U6 promoter and an indicated tRNA synthetase-encoding gene behind a CMV promoter. The construction of the plasmid was completed by replacing an AzFRS-encoding gene in pcDNA3.1-AzFRS[51] with the desirable DNA sequence. Fragments encoding the wild-type EcTyrRS or the evolved sTyrRS were amplified from E. coli genomic DNA or isolated library plasmid. Purified PCR product was assembled into a 6.2-kb DNA fragment from the digestion of pcDNA3.1-AzFRS by HindIII and ApaI.

**Plasmid pEGFP-double**. The 4.3 kb plasmid is derived from pEGFP[37] by replacing a 0.8 kb DNA fragment, which has flanking XbaI and ApaI sites, with a PCR product that contains the N150TAG mutation.

**Plasmid pCXCR4 and derivatives**. The 7.2-kb pCXCR4 plasmid is derived from pcDNA3.1 by replacing the AzFRS-encoding gene in pcDNA3.1-AzFRS with human CXCR4 variant 2. DNA fragment encoding human CXCR4 was amplified from pCEP4-FLAG-CXCR4[52] (Addgene 98947). Purified PCR product was assembled into a 6.2-kb DNA fragment from the digestion of pcDNA3.1-AzFRS by HindIII and ApaI. Mutations were introduced into the CXCR4 gene of pCXCR4 plasmid by mutagenic PCR.

**Plasmid pCXCR4-EGFP and derivatives**. The 8.0-kb pCXCR4-EGFP plasmid is derived from pcDNA3.1 by replacing the AzFRS-encoding gene in pcDNA3.1-AzFRS with a fusion gene between the human CXCR4 variant 2 and EGFP. DNA fragments encoding human CXCR4 without the stop codon and mammalian EGFP were amplified from plasmid pCXCR4 and pEGFP, respectively. Purified PCR products were assembled into a 6.2-kb DNA fragment from the digestion of pcDNA3.1-AzFRS by HindIII and ApaI. Mutations were introduced into the CXCR4 gene of pCXCR4-EGFP plasmid by mutagenic PCR.

**Plasmid pCXCR4(N38)-EGFP and derivatives**. The 7.0-kb pCXCR4(N38)-EGFP plasmid was constructed in a similar way to pCXCR4-EGFP, except the PCR product of DNA that encodes the first 38 amino acid of CXCR4 (CXCR4(N38)) was used in the Gibson assembly reaction. Obtained plasmid expressed a fusion protein CXCR4(N38)-EGFP. Mutations were introduced into the CXCR4(N38) sequence of pCXCR4(N38)-EGFP plasmid by mutagenic PCR.

**Plasmid pET-6xHis-SUMO-sTyrRS**. The plasmid was derived from a protein expression vector, pET-6xHis-SUMO, which was constructed in Dr. Limei Zhang's lab. DNA sequence encoding residues 1–322 of the evolved sTyrRS was amplified from psTyrRS. Purified PCR product was assembled into pET-6His-SUMO vector that was linearized by BamHI and XhoI.

**Library construction and selection**. To construct the $Ec$TyrRS library, four sites (Y37, L71, W129, and D182) were targeted for randomization using the NNK codon (N = A, C, T, or G, K = T or G) by mutagenic PCR. Four DNA fragments, each contained a single mutation site, were amplified using primers (Supplementary Table 1). Purified PCR products were assembled by overlapping PCRs[53] to produce a single DNA fragment that contained a 64-bp arm on each end that is homologous to the cloning vector, p$Ec$TyrRS-lib[32], which was linearized by digestion with EcoRI and BamHI. The $Ec$TyrRS library was generated by yeast homologous cloning method[54] as follows. To prepare electrocompetent yeast cells, single colony of MaV203/pGADGAL4 maintained on SD-Leu agar plate was used to start a seed culture, which was introduced into 100 mL fresh YPD media to reach a starting $OD_{600}$ of 0.3. Following cultivation at 30 °C for 6 h, cells were collected, washed, then resuspended in electroporation buffer(1 M sorbitol and 1 mM $CaCl_2$) to reach a final volume of 1 mL. The PCR product (30 μg) and linearized vector (10 μg) were mixed with the prepared competent cells. Electroporation was done at 2.5 kV and 25 μF in electroporation cuvettes (2 mm). Following electroporation, cells were recovered in 20 mL of sorbitol (1 M) and YPD mixture (1:1, v/v) at 30 °C for 1 h. Recovered cells were washed to remove extra nutrients, then cultured in 400 mL of SD-Leu-Trp media. A library size of $1 \times 10^7$ diversity was obtained.

Library selection was conducted by adapting a previously established procedure (Supplementary Fig. 1c)[32] as follows. First, a positive selection was conducted by culturing cells on plates without uracil to identify $Ec$TyrRS variants that can efficiently aminoacylate the amber suppressor tRNA with sTyr. A total of $1 \times 10^7$ cells was applied to the SD-Leu-Trp-Ura plates supplemented with 1 mM sTyr. Following incubation at 30 °C for 5 days, ~1000 colonies formed. These colonies were collected from the agar surface using SD-Leu-Trp media, then washed three times prior to the negative selection. In the negative selection, cells were plated on SD-Leu-Trp media plates containing 5-fluoroorotic acid (5-FOA) at 0.1%, but no sTyr. $Ec$TyrRS variants that can charge the amber suppressor $Ec$tRNA$_{CUA}$ with any one of the 20 natural amino acids led to GAL4-activated expression of URA3, which converts 5-FOA into a cytotoxic product and causes cell death. A total of $1 \times 10^4$ cells was subjected to the negative selection, and around 100 colonies formed after incubation at 30 °C for three days. Single colonies picked from the selection plates were resuspended in 100 μL of SD-Leu-Trp-Ura media. A 2 μL cell resuspension was applied to SD-Leu-Trp-Ura, SD-Leu-Trp-Ura+1 mM sTyr, and SD-Leu-Trp+0.1% FOA plates to further verify the phenotypes.

**Fluorescence analysis of yeast culture**. Yeast MaV203 strain harboring plasmid pyeGFP-N149TAG and a plasmid that encoded a sTyrRS variant was cultured in SD-Leu-Trp media without or with sTyr (1 or 5 mM) at 30 °C for 12 h following a 5% inoculation from a seed culture. Cells were then collected, washed, and then resuspended in PBS (pH 7.4). Cell density was determined by measuring $OD_{600nm}$. The fluorescence of yeGFP was monitored at $\lambda_{Ex}$ = 485 nm and $\lambda_{Em}$ = 528 nm.

Values of fluorescence intensity were normalized to cell growth. Reported data are the average of three measurements with standard deviations.

**Confocal microscopy**. 293T cells ($1 \times 10^5$) seeded in a single well of a 24-well plate were grown for 24 h, then transfected with plasmids psTyrRS (0.8 μg) and pEGFP (0.8 μg) in 2 μL of Lipofectamine. Transfected cells were cultured for an additional 24 h in 0.5 mL of media without or with sTyr (1 mM). Cells were then washed with 0.5 mL of warm DMEM base medium, fixed with 4% paraformaldehyde (w/v) for 15 min. Following removal of the fixation reagent by washing with $3 \times 0.5$ mL DPBS, cells were visualized by an Inverted (Olympus IX 81) confocal microscope.

**Flow cytometry analysis**. For the quantification of amber suppression efficiency by sTyrRS, 293T cells ($1 \times 10^5$) seeded in a single well of a 24-well plate were grown for 24 h, then transfected with one plasmid that encodes a tRNA synthetase (0.8 μg of psTyrRS, AZFRS, or $Ec$TyrRS) and pEGFP (0.8 μg) in 2 μL of Lipofectamine. Transfected cells were cultured for an additional 24 h in 0.5 mL of media without or with ncAA (1 mM of sTyr or AzF). Cells were detached with 0.3 mL of Trypsin/EDTA (0.05%, Thermo Fisher Scientific), washed once with 1 mL DPBS, and collected by centrifugation at $300 \times g$ for 5 min. Collected cells were resuspended in 0.5 mL 4% paraformaldehyde (w/v) and incubated for 20 min at room temperature. Following removal of the fixation reagent by centrifugation, cells were resuspended in 0.5 mL DPBS and kept on ice until analysis. Fluorescence of cells was measured using a Beckman Coulter CytoFLEX flow cytometer. A total of 30,000 cells were analyzed for each sample. Data were analyzed using FlowJo. Reported data are the average measurement of three samples with standard deviations. Flow cytometry analysis of sTyr incorporation into CXCR4(21TAG)-EGFP followed the similar method.

**In vitro tRNA synthetase assay**. For protein purification, plasmid pET30b-sTyrRS was transformed into protein expression host E. coli BL21(DE3). A 1 mL overnight seed culture was introduced into 100 mL of fresh LB medium containing kanamycin. When cell growth reached exponential phase ($OD_{600} \sim 0.4$–0.6), protein expression was induced with IPTG (0.5 mM). Cells were cultured continuously at room temperature for an additional 12 h prior to harvesting. Collected cells were lysed by sonication and tRNA synthetase with C-terminal 6xHis tag was purified on Ni resin by following the manufacturer's protocol, except for replacing phosphate buffer with Tris-HCl buffer (25 mM, pH 7.5). Protein concentration was measured by Bradford assay and purity was analyzed by SDS-PAGE.

For tRNA substrate preparation, the E. coli tyrosyl-tRNA substrate with CUA anticodon loop ($Ec$tRNA$_{CUA}$) was prepared by in vitro transcription using the MEGAscript® Kit (Life Technologies). A T7 promoter sequence (5′-TAATACGACTCACTATAGGG) was added to the 5′ end of the forward primer. Primers T7-Tyr-F and T7-Tyr-R were used to amplify the $Ec$tRNA$_{CUA}$ from p$Ec$TyrRS-lib template. A 20 μL reaction was set up with 0.2 μg PCR product, 8 μL NTPs mixture (18.75 mM each of ATP, CTP, GTP, and UTP), 2 μL enzyme mix, and 2 μL reaction buffer. The reaction was incubated at 37 °C for 12 h. The in vitro transcription was stopped by the addition of 150 μL ammonium acetate stop solution, then extracted with 150 μL phenol:chloroform:isoamyl alcohol (25:24:1). Collected aqueous phase was precipitated with an equal volume of isopropanol. Obtained tRNA pellet was dissolved in 50 μL nuclease-free water and the concentration was determined by $A_{260}$. The tRNA was folded prior to aminoacylation reactions by heating at 70 °C for 10 min, followed by the addition of $MgCl_2$ to 10 mM, then slow cooling by incubation at room temperature for 5 min.

For tRNA synthetase assay, standard in vitro aminoacylation assays were performed using EnzChek® Pyrophosphate Assay Kit (Thermo Fisher Scientific) to continuously monitor the release of PP$_i$[36,55]. The assay mixture contained ATP (0.1 mM), tRNA (10 μM), tRNA synthetase (5 μM), tyrosine or sulfotyrosine at specified concentrations, 2-amino-6-mercapto-7-methylpurine ribonucleoside (MESG, 0.2 mM), purine nucleoside phosphorylase (1 U/mL), and inorganic pyrophosphatase (0.03 U/mL) in a reaction buffer of HEPES (50 mM, pH 7.5) with $MgCl_2$ (10 mM), KCl (50 mM), and dithiothreitol (1 mM). Assays were carried out in 384-well format at room temperature by monitoring the absorbance change at 360 nm. Each reaction was initiated by the addition of an amino acid substrate. Control reactions without a substrate were performed. The standard curve of $A_{360nm}$ vs. PP$_i$ concentrations was constructed and used for the calculation of tRNA synthetase activities.

**Mass spectrometry analysis of sTyr incorporation in mammalian cells**. For protein purification, a T-75 flask was seeded with $3.0 \times 10^6$ 293T cells. Following 24 h of growth, cells were transfected with psTyrRS (19.2 μg) and pEGFP (19.2 μg) in 48 μL of Lipofectamine. Cells were cultured for an additional 24 h in media containing 1 mM of sTyr-D$_2$/sTyr mixture. Protein EGFP-Tyr40sTyr was purified from collected cells using Ni resin and further resolved from impurity by SDS-PAGE (Supplementary Fig. 4b). The band of EGFP-Tyr40sTyr was excised, washed, and treated with trypsin overnight at 37 °C. Tryptic peptides were extracted from the gel pieces, dried down, and re-dissolved in 25 μL of aqueous solution of acetonitrile (2.5%) and formic acid (0.1%).

For mass spectrometry analysis, all experiments were conducted using a Synapt G2S HDMS Q-TOF hybrid mass spectrometer (Waters, Manchester, UK) equipped with a nanoAcquity Ultra Performance LC (Waters, Manchester, UK) and an Acquity UPLC M-Class Peptide BEH C18 Column. Separation of peptides was performed using an acetonitrile gradient (0-85% ACN over 35 minutes) in the presence of 0.1% formic acid with the flow rate of 6 μL/min. The spectral acquisition was conducted in MassLynx 4.1 (Waters, Manchester, UK) with data handling and integration performed using IGOR Pro 6.3 (WaveMetrics, Lake Oswego, OR, USA). Deuterated percentages were calculated using peak area integrations of the $M_H$ and $[(M_H + 2) + M_D]$ peaks and predicted isotope abundances of the target molecules.

$$M_H * (\text{predicted isotope abundance}) - [(M_H + 2) + M_D] = M_D \text{ peak area} \quad (1)$$

$$\frac{M_D}{M_H} * 100 = \text{Deuterated Percentage} \quad (2)$$

**Crystallization of sTyrRS.** For protein purification, plasmid pET-6xHis-SUMO-sTyrRS was transformed into protein expression host *E. coli* BL21(DE3). Cell culturing, induction of protein expression, and purification of the tagged protein followed the same protocol as the purification of full-length sTyrRS. The N-terminal 6xHis-SUMO tag was removed by treatment with Ulp1. Following protease digest, the mixture was passed through Ni resin and the flow-through fraction, which contained sTyrRS (1–322), was collected. Obtained protein was dialyzed into the desalting buffer that contained NaCl (50 mM) and 2-mercaptoethanol (5 mM) in Tris-HCl buffer (20 mM, pH 7.5) using 10DG desalting columns (Bio-Rad Laboratories). Protein was further purified by gel filtration using a Superdex 200 column (GE Healthcare Life Sciences) at a flow rate of 2 mL/min. Collected fractions were concentrated to 10 mg/mL, flash frozen in liquid nitrogen, then stored at −80 °C until crystallization trials.

The sTyrRS$_{1-322}$—sulfotyrosine complex was crystalized by sitting drop vapor diffusion at 15 °C. Sulfotyrosine was added to the purified sTyrRS protein (a.a. 1–322, 9 mg/mL) to a final concentration of 2 mM. Obtained protein solution was then mixed at a 2:1 ratio (v/v) with the reservoir solution containing ammonium sulfate (2.2 M) and PEG 400 (2.5–3%, v/v) in HEPES buffer (0.1 M, pH 7.5). The crystals typically reached the maximum size by three days. Fomblin® Y LVAC 16/6 (MilliporeSigma) was used as a cryoprotectant during crystal harvesting.

The X-Ray diffraction data were collected at Beamline 12-2 of the Stanford Synchrotron Radiation Lightsource (SSRL), with a 6 M Pixel Array Detector. A set of 1440 diffraction images were collected at 12660.4 eV with an oscillation angle of 0.25°. Diffraction data were integrated with the XDS program and subsequently processed using the CCP4 suite[56,57]. The crystal structure of *Escherichia coli* TyrRS (PDB ID: 2YXN) was used as the initial model for phase determination[58]. Further structural refinement was carried out using PHENIX, REFMAC5 and COOT[59–61]. The structural validation was performed by MolProbity[62]. The PyMol (Schrödinger Inc.) program was used for structural analysis and for generating figures. The diffraction data and refinement statistics are summarized in Supplementary Table 3. The 2Fc-Fo electron density map (3.0 σ) and Fo-Fc omit electron density map (3.0 σ) are shown in Supplementary Fig. 6b and 6c. The atomic coordinates and experimental data (code 6WN2) have been deposited in the Protein Data Bank (www.wwpdb.org).

**Generation of CXCR4-knockout cell line.** A CXCR4-knockout cell line was generated from 29T using CRISPR/Cas9 method[43]. The sequence of the guide RNA (5′-ACTTCAGATAACTACACCG-3′) was selected using an online service provided by Integrated DNA Technologies. It targeted the 5′-end of the second exon of the CXCR4. The gRNA was inserted into BbsI-linearized plasmid pSpCas9(BB)-2A-GFP[43] (PX458, Addgene 48138). Obtained plasmid was transfected into 293T cells. Following cultivation for 24 h, cells were detached and single cells with GFP expression were sorted into a 96 well plate using BD FACSAria II. After cell expansion for two weeks, genome editing events in single cell-derived colonies were examined by T7 endonuclease I assay (T7E1)[63]. A 0.7 kb region flanking the site of potential chromosomal editing was amplified by PCR. The PCR product was purified, then mixed at 1:1 ratio with PCR product of 293T cell line. The DNA mixture was subjected to an annealing reaction to promote the formation of heteroduplex[63]. Obtained products were treated by T7 endonuclease I and analyzed by DNA gel electrophoresis. PCR products that led to positive digestion results were subjected to DNA sequencing analysis to identity cell clones with potential homozygous editing events. The CXCR4 region of two candidates was further amplified and ligated into pUC19 vector. Ten bacterial clones from each ligation were sequenced to further verify the genotype. The selected 293T CXCR4-knockout cell line had an insertion of A at position 7 of the second exon on both chromosomes, which led to an early termination of CXCR4 translation. The phenotype of the cell line was confirmed by calcium mobilization assays.

**Calcium mobilization assay.** Cells cultured to 90% confluency were detached then seeded in a single well ($1 \times 10^5$ cells) of a poly-lysine coated, clear bottom, black wall 96 well plate (Greiner Bio-One). Following incubation for 24 h at 37 °C, the culture media was replaced with 100 μL DMEM containing 4 μM Fluo-4 AM (Thermo Fisher Scientific) and 0.04% pluronic acid. Following an hour incubation

at room temperature protected from light, media was replaced with fresh DMEM and cells were incubated at room temperature protected from light for another 45 min. Cells were washed with 100 μL DPBS once, then immersed in 100 μL fresh DPBS and ready for assay. Calcium mobilization was measured using excitation at 485 nm and emission at 528 nm. Fluorescence of cells was monitor for 2 min prior to the addition of SDF-1 (20 nM) and for 4 min after the stimulation. The calcium response was calculated as the percentage of fluorescence change over baseline. Data are expressed as the average of three measurements with standard deviation.

**Statistics and reproducibility.** Standard deviations and Student's *t* test were performed using Microsoft Excel 365 software. Similar results were obtained from three independent experiments.

**Reporting summary.** Further information on research design is available in the Nature Research Reporting Summary linked to this article.

## Data availability

All data generated or analyzed during this study are included in this article and its Supplementary Information. All relevant data are available from the authors upon reasonable request. Source data are provided with this paper. Crystal structure of sTyrRS in complex with sTyr is deposited in Protein Data Bank (PDB ID: 6WN2. Crystal structures of *E. coli* tyrosyl-tRNA synthetase (TyrRS) in complex with L-tyrosine (PDB ID: 1X8X) and *E. coli* TyrRS (PDB ID: 2YXN) are accessed from Protein Data Bank. Source data are provided with this paper.

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

## Acknowledgements

The authors thank Dr. You Zhou and Ms. Terri Fangman (Morrison Microscopy Core Research Facility at University of Nebraska-Lincoln) for assistance in confocal microscope analysis, Mr. Dirk Anderson (Flow Cytometry Service Center at University of Nebraska-Lincoln) for assistance in flow cytometry analysis. The authors thank staff at Beamline 12-2 of Stanford Synchrotron Radiation Lightsource for the assistance during the X-ray diffraction data collection. Stanford Synchrotron Radiation Lightsource, SLAC National Accelerator Laboratory, is supported by the U.S. Department of Energy, Office of Science, Office of Basic Energy Sciences [DE-AC02-76SF00515]. This work was supported by New Faculty Startup Fund from University of Nebraska-Lincoln (to W.N.), National Science Foundation (grant CBET 1805528 to W.N. and J.G.), National Science Foundation (grant MCB 1553041 to J.G.), National Institute of Health (grant 1R01AI111862 to J.G. and W.N.), National Science Foundation (grant CLP 1846908 to L-M.Z.), National Institute of Health (grant P30 GM103335 to L-M.Z. through the Nebraska Redox Biology Center), and MacNair Scholarship from University of Nebraska-Lincoln (to D.G.B.).

## Author contributions

W.N. and J.G. designed the study. X.H., Y.C., and W.N. constructed plasmids. X.H. constructed the *Ec*TyrRS library and performed selection in yeast, purified protein and conducted in vitro enzyme assays, expressed and purified protein for M.S. analysis, expressed and purified protein for crystallization study. X.H. and Y.C. conducted confocal microscopy and flow cytometry studies. J.L. and M.K. performed mass spectrometry studies. D.G.B. conducted protein crystallization and data collection. B.M. synthesized sulfotyrosine. M.K. and E.D. analyzed mass spectrometry data. D.G.B. and L-M.Z. solved and refined protein structure. X.H. and W.N. constructed 293T CXCR4-knockout cell line and conducted calcium mobilization assay. F.W. helped design the *Ec*TyrRS mutant library. X.H., J.G., and W.N. analyzed the data and wrote the manuscript.

## Competing interests

The authors declare no competing interests.

**Additional information**

