## [Peer Review File · Nature Communications]

Reviewers' Comments:

Reviewer #1:

Remarks to the Author:

Wei Niu's group reports the site-specific sulfation of proteins by introducing L-sulfotyrosine (sTyr) as a 21st amino acid into proteins in mammalian cells. To this end, they engineered E. coli tyrosyl-tRNA synthetase (EcTyrRS) as the orthogonal pair with amber suppressor E. coli tRNA for the sTyr aminoacylation. Another report by Chatterjee's group with the very similar method has been published in Nat. Chem. Biol. Chatterjee's group developed the engineered EcTyrRS by evolving the RS/tRNA pair in E. coli using a library of the RS mutants randomized six active site residues. In contrast, Niu's group used a yeast system for the evolution with a library of the mutants randomized four active site residues. Both teams succeeded in the development of the orthogonal RS/tRNA pair for the sTyr incorporation into proteins in mammalian cells. In addition, Niu's group analyzed the crystal structure of the evolved EcTyrRS variant in complex with sTyr and examined the biological activity of chemokine receptor CXCR4, in which sTyr was site-specifically incorporated into CXCR4 in mammalian cells using the RS/tRNA pair. The research is solid and would draw attention in combination with the Chatterjee's report. However, many readers might be interested in the results and discussion about the following comments 1 and 2.

1. The authors' mutant with high specificity contains L71V, W129F, and D182G. In contrast, the Chatterjee's mutant contains L71V, D182G, and F186M. The audience of the journal might want to know the authors' discussion including the Chatterjee's mutant, based on their structural analysis. In addition to the F186M explanation, Supplementary Fig. 5 should be added to Fig. 3 in the manuscript and modified, to discuss the four mutant sites at positions 71, 129, 182, and 186.

2. Although the calcium response of CXCR4(FF21TAG) is as high as that of wt, the audience might want to know the functions of the other CXCR4 mutants at three positions of Y7, Y12, and Y21, including multiple incorporation of sTyr. Chatterjee's group expressed human heparin cofactor II (HCII) containing two sTyrs. Alternatively, the authors should discuss the multiple sTyr incorporation if the multiple incorporation significantly reduced the expression of CXCR4 mutants.

3. The authors should explain why they used a mixture of 24.2% the di-deuterium derivative of sTyr (sTyr-D2) with sTyr, for MS analysis.

4. Lines 246-247 on page 11, the authors might explain why the Tyrs at positions 7 and 12 were mutated to Phe, for audience to easier understand.

5. The following typos in the references should be fixed.

Ref 4: New Biotechnol. > N. Biotechnol.

Ref 6: Von, F. K. -> von Figura, K. (Kurt von FIGURA) ?

Ref 30: e36/31-e36/38 -> e36

Ref 46: Mol. BioSyst. -> Mol. Biosyst.

Ref 47: Nature methods 6, 343 -> Nat. Methods 6, 343-345.

Ref 49: Science (Washington, DC. U.S.) -> Science

Ref 53: Protein engineering, design & selection: PEDS -> Protein Eng. Des. Sel.

Ref 54: Journal of biomolecular screening -> J. Biomol. Screen.

Ref62: 69/61-69/11 -> 69

Reviewer #2:

Remarks to the Author:

The manuscript describes the genetic encoding of sulfotyrosine in mammalian cells. The importance of this PTM is unclear for biology as of yet we do not have effective tools to probe its role. The recent paper by Chatterjee describe the nearly identical work presented here. While there are slightly different approaches the general outcome is nearly identical. Both systems function as reported. A comparison to the Chatterjee manuscript is provided at the end but worth to note is that the authors used a different selection method and different starting library but resulted in nearly identical synthetases. The value of the contribution of this work to the field is the same as that of the Chatterjee work but with the the addition of structural characterization and a different biological demonstration of function. I recommend this manuscript be published as a companion piece with the Chatterjee manuscript. From a GCE and chemical biology perspective the papers strengthen each other.

Major problems.

There is structural interpretation clarity that needs to be improved.

- 1) Authors ***must*** show a 2Fo-Fc map (> 1 sigma) for the sulfo-tyrosine moiety to confirm accurate modeling of substrate in the active site the synthetase.
- 2) Bond angles of the sulfo-Tyr seems highly skewed in several places, which can be seen quite readily in Supporting Fig. 5b. Did the author's have to adjust standard restraints to make this possible? If so, this should be included in the methods section and commented on in the text. A figure confirming accurate modeling the skewed substrate confirmation should be possible with the reported resolution, and should be shown, e.g. showing 2Fo-Fc density contoured high (>3 sigma) to show accurate positioning of each atom.
- 3) Confirmation that the WT CXCR5 protein is sulfated, and that the FF-Y21TAG variant is also sulfated to a similar degree would contextualize the conclusions regarding sulfation at site 21 being important to calcium mobilization and make the result much more relevant.

Minor problems.

- 4) Supporting Fig. 5C is a poor representation of the relative positioning of phenolic rings of sulfo-Tyr vs Tyr. A zoomed in perspective would provide a more appropriate comparison.
- 5) The statement (page 10 of main text), "The three hydrogen bonds that are essential for recognizing the α -amino group of the substrate in the wild-type EcTyrRS are preserved in the evolved sTyrRS" seems odd considering those amino acids were not included in the library. Was there any other alternative?
- 6) Visual representation of the active sites in Fig. 3 of main text would benefit from improvement. Removal of backbone interactions (which were conserved) and a better focus on the side chain interactions would help. Remove Gln179, Gln201, Tyr175 and Asp81 since they impart no functional recognition to the side chain. Instead, a zoomed in

view of the sulfo-group interactions (including waters) should be made. A visual representation of the Tyr37  sulfo group hydrogen bond would help since the authors make reference to this in the main text yet from the current representation it would appear is an unfavorable angle of interaction.

- 7) The library says it was quality-checked by sequencing, but no evidence of this is reported. The authors state that their library of 1×10^7 members covers 99% of the theoretical library. There should be a reference associated with this claim. I recommend the following: <https://pubmed.ncbi.nlm.nih.gov/15857784/>
- 8) On page 6, lines 129-130, the authors state: "Such modification was shown to have negligible effect on the function of the wild-type *EcTyrRS*.³⁴" This wording is a bit confusing to me, as it implies that the authors explicitly checked to see that this modification affected activity; however, no results are shown to support this. If, however, the authors mean to state that the modification *has been shown* (as would be implied by the citation) to have negligible effect, the wording should make this clear.
- 9) The MS analysis seems confusing to me such that I'm not sure I understand it fully. MS was used, as stated, to quantify fidelity; however, "The data excluded misincorporation of unlabeled Tyr in the culture media as a source of Tyr at position 40." How can any statements regarding fidelity, such as "Overall, above studies demonstrate that the evolved sTyrRS can aminoacylate amber suppresser tRNA with high efficiency and fidelity in mammalian cells" be made then?

Thus, to me, the data is essentially saying that when sTyr is added to the media with a specific proportion deuterated, purified protein that contains sTyr also has a similar deuterium ratio. This is basically saying that the same sTyr that was added to the media is the same sTyr that has been incorporated, when one ignores Tyr incorporation. This does not address sTyr hydrolysis and incorporation as Tyr and is also inconsistent with previously reported evidence that Tyr can be incorporated (e.g. Fig. 1 and Fig. 2).

Please clarify the value of the deuterated MS work.

- 10) Unless there is a typo on page 9, lines 187-189 "As anticipated, although sTyr was observed at position 40, the signal strength was very low (data not shown)."
 1. Where "sTyr" should have been "Tyr"?
 2. This is consistent with a statement made in the discussion, page 13, lines 296-298 "However, mis-incorporation of Tyr was not observed in the presence of sTyr, which suggests that the evolved sTyrRS is more active on sTyr than Tyr."

Otherwise, I see no support for this statement, as Tyr incorporation is not discussed in the MS section. Please clarify.

- 11) It would have been nice to see the effects of the other sTyr sites in CXCR4, but this is not required for the current work
 - i. An explanation for why would be good:
 1. Was the system not capable of incorporating at these sites?
 2. Were these sites not considered relevant?

	Work	Chatterjee	Niu
Selection	aaRS	EcTyrRS	EcTyrRS
	Sites (Common)	Y37: (FLIMVSTAYHCG) L71: NBT N126: (NSPTACGDH) D182: NST F183: NNK L186: NNK	Y37: NNK L71: NNK W129: NNK D182: NNK
	Selection Host	Escherichia coli (ATMY3)	Saccharomyces cerevisiae (MaV230)
	Selection Architecture	Standard Double Sieve (single round)	1 round positive (sTyr) 1 round negative (5-FOA)
	Hits (Selected)	Hit 1: L71V+D182G Hit 2: L71V+D182G+L186M	Hit 1: L71V+D182G Hit 2: L71V+W129F+D182G
Characterization	Suppression Efficiency	Measured in E. coli	Measured in S. cerevisiae /293T
	Fidelity	Measured in vivo (E. coli /293T) and by mass spectrometry (E. coli /293T)	Measured in vivo (S. cerevisiae /293T) / in vitro / by mass spectrometry? (293T)
	Incorporation Confirmation	Detected by MS and western blot (sfGFP from E. coli /293T cells); MS on HCII from 293T	Detected by MS (293T) and by western blot (CXCR4 from HEK293T)
	Structural Analysis	-	Determined by X-Ray Crystallography Analysis
	Mammalian	Used microscopy/fluorescence of lysate to observe incorporation	Used microscopy/FACS to observe incorporation
	Experiments	1. Used fluorescent reporter (sfGFP-Y149TAG) in E. coli to observe suppression efficiency 2. Performed MS and WB on sfGFP from E. coli (anti-sTyr) 3. Used fluorescent reporter (EGFP-Y39TAG) to observe suppression efficiency in 293T cells by microscopy and in clarified cell lysate 4. Performed MS and WB on EGFP from 293T cells (anti-sTyr) 5. Performed MS on HCII expressed in 293T cells	1. Used a fluorescent reporter (yeGFP-N149TAG) in S. cerevisiae to observe suppression efficiency 2. Performed in vitro pyrophosphate release assay (sTyr and Tyr as substrates) 3. Observed sTyr suppression efficiency in 293T by microscopy and FACS 4. Performed mass spectrometry analysis (purified protein from HEK293T) 5. Determined sTyrRS structure with sTyr bound by X-Ray crystallography 6. Checked incorporation by of sTyr in CXCR4 by western blotting (anti-sTyr)
Application	Protein (sites)	HCII (S60TAG, S73TAG)	CXCR4 (Y21TAG)
	Expression Host	HEK293T	HEK293T (Δ CXCR4)
	Assay	Thrombin inhibition assay	Calcium mobilization assay
	Result	Y60[sTyr] > Y72[sTyr] for thrombin inhibition	Y21[sTyr] is important for SDF-1 binding by CXCR4

Response to reviewer comments

Reviewer #1

1. The authors' mutant with high specificity contains L71V, W129F, and D182G. In contrast, the Chatterjee's mutant contains L71V, D182G, and F186M. The audience of the journal might want to know the authors' discussion including the Chatterjee's mutant, based on their structural analysis. In addition to the F186M explanation, Supplementary Fig. 5 should be added to Fig. 3 in the manuscript and modified, to discuss the four mutant sites at positions 71, 129, 182, and 186.

We thank the reviewer for these constructive suggestions. The key items from original Supplementary Fig. 5 have been added to Fig. 3 in the revised manuscript. A discussion on the L186M mutation is included in the "Discussion" section of the revised manuscript.

2. Although the calcium response of CXCR4(FF21TAG) is as high as that of wt, the audience might want to know the functions of the other CXCR4 mutants at three positions of Y7, Y12, and Y21, including multiple incorporation of sTyr. Chatterjee's group expressed human heparin cofactor II (HCII) containing two sTyrs. Alternatively, the authors should discuss the multiple sTyr incorporation if the multiple incorporation significantly reduced the expression of CXCR4 mutants.

While a side-by-side evaluation has not been conducted with evolved sTyrRSs from this and Chatterjee's work, comparable activities are expected since they have similar mutations, including Leu71Val and Asp182Gly. We added a few sentences to describe this notion in the Discussion section of the manuscript. In addition, we conducted protein expression of an EGFP mutant that contains two amber codons. The data is shown in Supplementary Fig. 3 and the description of this experiment has been added to the main text.

The main goal of this manuscript is to demonstrate the genetic encoding of sTyr. The purpose of our functional study of an essential sulfation site (i.e. sTyr21) in CXCR4 was to show that the genetic encoding approach could be applied to cellular studies in live mammalian cells. A complete study of all sulfation sites is beyond the scope of this work. We plan to conduct such studies and will report results in the future.

3. The authors should explain why they used a mixture of 24.2% the di-deuterium derivative of sTyr (sTyr-D2) with sTyr, for MS analysis.

The reviewer's suggestion is well received. Due to the cost of the deuterium labeled reagent, we decided to use a mixture of labeled and un-labeled sTyr for mass spectrometry study of purified protein. Our intention was to have about 25% deuterium labeled sTyr in the mixture, which should be sufficient for the mass spectrometry analysis. The actual percentage of deuterium labeled sTyr was determined to be $24.2 \pm 0.3\%$ (mol%) by mass spectrometry quantification. We have added a brief description and discussion in the manuscript.

4. Lines 246-247 on page 11, the authors might explain why the Tyrs at positions 7 and 12 were mutated to Phes, for audience to easier understand.

We thank the reviewer for the constructive suggestions. We have added a brief description in the revised manuscript.

5. The following typos in the references should be fixed.
Ref 4: *New Biotechnol.* > *N. Biotechnol.*
Ref 6: Von, F. K. -> von Figura, K. (Kurt von FIGURA) ?
Ref 30: e36/31-e36/38 -> e36
Ref 46: *Mol. BioSyst.* -> *Mol. Biosyst.*
Ref 47: *Nature methods* 6, 343 -> *Nat. Methods* 6, 343-345.
Ref 49: *Science (Washington, DC. U.S.)* -> *Science*
Ref 53: *Protein engineering, design & selection: PEDS* -> *Protein Eng. Des. Sel.*
Ref 54: *Journal of biomolecular screening* -> *J. Biomol. Screen.*
Ref62: 69/61-69/11 -> 69

We thank the reviewer for identifying errors in references. We have made corresponding changes in the revised manuscript.

Reviewer #2

1) Authors must show a 2Fo-Fc map (> 1 sigma) for the sulfo-tyrosine moiety to confirm accurate modeling of substrate in the active site the synthetase.

The reviewer's suggestion is well received. We have added two electron density maps to the revised manuscript. (1) Supplementary Fig. 6b is a 2Fo-Fc map (3σ) with focus on sulfotyrosine, Y37, and N126. (2) Supplementary Fig. 6c is a Fo-Fc omit electron density map (3σ) around the sTyr site of the sTyrRS complexed with sTyr.

2) Bond angles of the sulfo-Tyr seems highly skewed in several places, which can be seen quite readily in Supporting Fig. 5b. Did the author's have to adjust standard restraints to make this possible? If so, this should be included in the methods section and commented on in the text. A figure confirming accurate modeling the skewed substrate confirmation should be possible with the reported resolution, and should be shown, e.g. showing 2Fo-Fc density contoured high (>3 sigma) to show accurate positioning of each atom.

We appreciate the reviewer's comments. Standard restraints were used for all refinements with the built-in ligand library. We provided additional electron density maps (Supplementary Fig. 6b and 6c) to support the modeling of the sulfotyrosine ligand. The observed skewed bond angles in the original Supplementary Fig. 5b of the submitted manuscript could be caused by poor representation of the structure. We re-graph the Figure and the new one is now included in the main text of the revised manuscript as Fig. 3a.

3) Confirmation that the WT CXCR5 protein is sulfated, and that the FF-Y21TAG variant is also sulfated to a similar degree would contextualize the conclusions regarding sulfation at site 21 being important to calcium mobilization and make the result much more relevant.

The reviewer's comment is well received. Due to the very labile nature of the sulfoester bond in sTyr under mass spectrometry conditions, it is not possible to quantify the sulfation degrees by mass spectrometry. As an alternative approach, we conducted western blot experiments using anti-sulfotyrosin antibody. While we spent over five weeks and tried multiple different protocols, the difficulty of purifying CXCR4 as an intact membrane protein prevented us from obtaining conclusive data on the relative sulfation degrees of the wild-type and mutant CXCR4.

Since the importance of sTyr21 in CXCR4 function has been demonstrated in literature, we feel that our current data can sufficiently serve the goal of this work, which is to show that the genetic encoding approach could be applied to cellular studies in live mammalian cells. We revised our discussion to clearly state that our data validate the previous finding that sTyr21 can support CXCR4 function.

4) Supporting Fig. 5C is a poor representation of the relative positioning of phenolic rings of sulfo-Tyr vs Tyr. A zoomed in perspective would provide a more appropriate comparison.

We revised the original Supporting Fig. 5C according to reviewer's suggestion and it is now presented as Supplementary Fig. 6f.

5) The statement (page 10 of main text), "The three hydrogen bonds that are essential for recognizing the α -amino group of the substrate in the wild-type EcTyrRS are preserved in the evolved sTyrRS" seems odd considering those amino acids were not included in the library. Was there any other alternative?

The reviewer is correct. We have revised the description to avoid confusions.

6) Visual representation of the active sites in Fig. 3 of main text would benefit from improvement. Removal of backbone interactions (which were conserved) and a better focus on the side chain interactions would help. Remove Gln179, Gln201, Tyr175 and Asp81 since they impart no functional recognition to the side chain. Instead, a zoomed in view of the sulfo-group interactions (including waters) should be made. A visual representation of the Tyr37  sulfo group hydrogen bond would help since the authors make reference to this in the main text yet from the current representation it would appear is an unfavorable angle of interaction.

We thank the reviewer for these valuable suggestions. We modified Fig. 3 in the original manuscript as following. (1) Figure 3c in the revised manuscript focuses on interactions of the sulfate group with the protein. Hydrogen bonds with Tyr37 and Asn126 residues are labelled. (2) Additional graphs that show the interactions between the amino group on the ligand and the protein are included as Supplementary Fig. 6d and 6e.

*7) The library says it was quality-checked by sequencing, but no evidence of this is reported. The authors state that their library of 1×10^7 members covers 99% of the theoretical library. There should be a reference associated with this claim. I recommend the following:
<https://pubmed.ncbi.nlm.nih.gov/15857784/>*

We have added sequencing data (Supplementary Fig. 1b) to support our claim. The suggested reference was also included in the revised manuscript (ref 31).

8) On page 6, lines 129-130, the authors state: "Such modification was shown to have negligible effect on the function of the wild-type EcTyrRS.³⁴" This wording is a bit confusing to me, as it implies that the authors explicitly checked to see that this modification affected activity; however, no results are shown to support this. If, however, the authors mean to state that the modification has been shown (as would be implied by the citation) to have negligible effect, the wording should make this clear

The reviewer is correct. This sentence was intended to describe a previously reported observation by Kiga *et al.* (ref 35). We have revised the description to avoid confusions.

9) The MS analysis seems confusing to me such that I'm not sure I understand it fully. MS was used, as stated, to quantify fidelity; however, "The data excluded misincorporation of unlabeled Tyr in the culture media as a source of Tyr at position 40." How can any statements regarding fidelity, such as "Overall, above studies demonstrate that the evolved sTyrRS can aminoacylate amber suppresser tRNA with high efficiency and fidelity in mammalian cells" be made then?

Thus, to me, the data is essentially saying that when sTyr is added to the media with a specific proportion deuterated, purified protein that contains sTyr also has a similar deuterium ratio. This is basically saying that the same sTyr that was added to the media is the same sTyr that has been incorporated, when one ignores Tyr incorporation. This does not address sTyr hydrolysis and incorporation as Tyr and is also inconsistent with previously reported evidence that Tyr can be incorporated (e.g. Fig. 1 and Fig. 2).

Please clarify the value of the deuterated MS work.

We agree with the reviewer that our initial description causes confusion. This question needs to be addressed together with the next question. Due to the labile nature of sulfoester bond under the MS condition, we only observed weak signals of sulfotyrosine-containing peptide. Therefore, our data analysis of the MS experiment focused on the peptide that contains Tyr at the targeted position 40 in EGFP. Possible sources of Tyr40-containing peptide include cleavage of the sulfate group in sTyr40-containing peptide under the MS condition and misincorporation of Tyr by the evolved sTyrRS. Since the cell culture media (DMEM with 10% FBS) contains over 0.4 mM Tyr (a component of DMEM; <https://www.sigmaaldrich.com/life-science/cell-culture/learning-center/media-formulations/dme.html>) and sTyr (25% deuterium label) was added to 1 mM, we should see a significant dilution of the deuterium label in this peptide if the evolved sTyrRS incorporates Tyr at a considerable level. However, we observed that the percentage of deuterium label is consistent between the peptide and the added sTyr. This observation led to our conclusion that the evolved sTyrRS has high fidelity. We have added a more detailed explanation in this regard to the revised manuscript.

10) Unless there is a typo on page 9, lines 187-189 "As anticipated, although sTyr was observed at position 40, the signal strength was very low (data not shown)."

1. Where "sTyr" should have been "Tyr"?

2. This is consistent with a statement made in the discussion, page 13, lines 296-298

"However, mis-incorporation of Tyr was not observed in the presence of sTyr, which suggests that the evolved sTyrRS is more active on sTyr than Tyr."

Otherwise, I see no support for this statement, as Tyr incorporation is not discussed in the MS section. Please clarify.

We agree with the reviewer that our initial description is not clear. Please also refer to explanations to the last question of our response.

1. The sentence in question does not contain a typo. Due to the well documented labile nature of the sulfoester bond in sTyr under mass spectrometry conditions, we expected to and did observe weak signal of the sTyr-containing target peptide.
2. The statement is based on the conclusion of the MS analysis, where we observed the preservation of deuterium label in the target peptide.

We have clarified this notion in the revised manuscript.

11) It would have been nice to see the effects of the other sTyr sites in CXCR4, but this is not required for the current work

i. An explanation for why would be good:

1. Was the system not capable of incorporating at these sites?

2. Were these sites not considered relevant?

According to previously reported studies, the relative importance of the three sulfation sites of CXCR4 for its ligand binding and activation is Tyr21 > Tyr12 > Tyr7. Therefore, we decided to focus on the Tyr21 site of CXCR4 in this work. We added a brief discussion in the revised manuscript.

As the reviewer pointed out, a complete functional study of all sulfation sites is beyond the scope of this work since our purpose is to show that the genetic encoding approach could be applied to cellular studies in live mammalian cells. We plan to conduct such studies and will report results in the future.

Reviewers' Comments:

Reviewer #1:

Remarks to the Author:

The authors have revised all of the parts according to the reviewers' comments. I think the current revised version is suitable for publication.